# Bone Loss, Osteoporosis, and Fractures in Patients with Rheumatoid Arthritis: A Review

**DOI:** 10.3390/jcm9103361

**Published:** 2020-10-20

**Authors:** Patrice Fardellone, Emad Salawati, Laure Le Monnier, Vincent Goëb

**Affiliations:** 1Department of Rheumatology, Picardie-Jules Verne University, University hospital of Amiens, 80054 Amiens, France; LeMonnier.Laure@chu-amiens.fr (L.L.M.); goeb.vincent@chu-amiens.fr (V.G.); 2Assistant Professor, Faculty of Medicine, King Abdulaziz University, 21589 Jeddah, Saudi Arabia; esalawati@kau.edu.sa

**Keywords:** rheumatoid arthritis, osteoporosis, fragility fractures, BMD, bone remodeling markers

## Abstract

Rheumatoid arthritis (RA) is often characterized by bone loss and fragility fractures and is a frequent comorbidity. Compared with a matched population, RA patients with fractures have more common risk factors of osteoporosis and fragility fractures but also risk factors resulting from the disease itself such as duration, intensity of the inflammation and disability, and cachexia. The inflammatory reaction in the synovium results in the production of numerous cytokines (interleukin-1, interleukin-6, tumor necrosis factor) that activate osteoclasts and mediate cartilage and bone destruction of the joints, but also have a systemic effect leading to generalized bone loss. Regular bone mineral density (BMD) measurement, fracture risk assessment using tools such as the FRAX algorithm, and vertebral fracture assessment (VFA) should be performed for early detection of osteoporosis and accurate treatment in RA patients.

## 1. Introduction

Rheumatoid arthritis is the most common immune disease of the skeleton. During the last three decades, many studies have shown that besides the joint inflammation and destruction, bone mass in patients with rheumatoid arthritis (RA) is lower than in the matched non-RA population [1,2,3,4,5,6,7,8,9] and the risk of osteoporosis is increased [8,10,11,12,13,14,15,16,17,18,19]. Similarly, the incidence of osteoporotic fractures is high [6,11,12,14,15,16,17,18,19,20,21,22,23,24,25] even when compared to the general population of the same age and sex [5,9,15,26,27,28,29,30,31,32,33,34,35,36,37,38,39,40,41,42]. Osteoporosis-related fragility fractures represent one of the most severe complications in RA patients [43]. They contribute to a significant decrease in the quality of life and can reduce life expectancy. In the United States, data from the National Data Bank for Rheumatic Diseases indicated that osteoporotic fractures are the third cause of mortality in RA patients [44]. The severe inflammatory reaction in the synovium results in the production of numerous cytokines (interleukin-1, interleukin-6, tumor necrosis factor) that mediate cartilage and bone destruction but also have a systemic effect leading to generalized bone loss. As in the general population, the highly sensitive C-reactive protein (CRP) level is a predictor of the risk of fracture that is underlying the role of systemic inflammation [45]. The mechanism of bone loss relies on increased bone resorption mediated by pro-inflammatory cytokines that activate osteoclasts, as well as by the frequent use of glucocorticoids or inactivity resulting from the disease. Some clues lowering the level of inflammation with potent drugs such as biologic agents can reduce the risk of osteoporosis and fractures. The literature provides interesting data on the prevalence of osteoporosis in RA patients, as well as the incidence of fragility fractures compared to a healthy population.

## 2. Epidemiology of Bone Loss and Fractures in Rheumatoid Patients

### 2.1. Low BMD

According to the definition given by the WHO organization in 1994, based on a bone mineral density (BMD) T-score ≤ 2.5 SD measured by dual-energy X-ray absorptiometry (DEXA) in the spine and/or hip, it is possible to evaluate the prevalence of osteoporosis in the RA population. In this population, decreased mean BMD values of the spine and hips are known to be a very common feature and are more frequent than in a paired healthy control population (Table 1). A significant bone loss appears early in RA. After a two-year duration of the disease, the hip and spine BMD show an already significant loss in all locations, with a significantly greater BMD loss in the hands (BMD measured in metacarpals 2–4 by digital x-ray radiogrammetry) than a generalized BMD loss in the hip and spine [46]. In an established RA, observational and case-control studies estimate the prevalence of loss of BMD to be doubled in both male and female RA patients compared with healthy controls at both the lumbar spine and the hip (Table 1 and Table 2). Results are expressed whether in T-score or Z-score. T-score is a young gender-matched population at peak bone mass, while the Z-score is derived from an age-matched reference population. Both T-scores and Z-scores are derived by comparison to the reference population on a standard deviation scale.

As shown in Table 1, among the classical risk factors of bone loss, glucocorticoids (GCs) commonly prescribed in RA patients are also well known for causing secondary osteoporosis with the long use of high dosage when comparing users with non-users [1,2,4]. Inversely, it seems that at low dosage, glucocorticoids exert a protective effect on the bone probably because the lowering of the inflammation is more beneficial than the deleterious effects of the drug [47].

### 2.2. Fragility Fractures

Epidemiological studies on the frequency of fractures in RA populations can be observational, case-controlled, conducted on databases, or prospective (Table 3 and Table 4). Most epidemiological studies provide a fractures risk increased by 1.5- to 2-fold among patients with RA compared to the general population (Table 4).

Patients with RA are at a high risk of vertebral fractures. However, the prevalence of spine fractures varies enormously according to the population studied (age, women alone, or men and women), the way to assess fractures (radiographs or vertebral fracture assessment (VFA) for vertebral fractures), and the source of data (cohort, registry, randomized trial). This explains that the prevalence of vertebral fractures ranges from 8% to nearly 50% (Table 3).

On the other hand, the risk of fractures, despite the site, is always higher in RA patients than in the general population (Table 4). In a very large case-control study of 30,262 patients with RA using the British General Practice Research Database, the increased risk of fracture compared to the general population was most marked at the hip (RR: 2.0, 95% CI 1.8–2.3) and spine (RR: 2.4, 95% CI 2.0–2.8) [32]. The relationship between RA and the fracture was examined in the Women’s Health Initiative prospective study (WHI). Compared to the non-arthritis group, the risk (HR (95% CI)) of sustaining any clinical fracture comprising self-reported spinal fractures in the RA group was 1.49 (1.26, 1.75) (*p* < 0.001). Moreover, the risk of sustaining a hip fracture significantly increased in the RA group (3.03 (2.03, 4.51)) (*p* < 0.001) [34].

Cohort and registry studies allow calculating the incidence-rates of fractures mostly for peripheral fractures. A Canadian nested case-control study conducted using Quebec physician billing, pharmacy data from the “Régie de l’assurance maladie du Québec”, and hospital discharge data found that the incidence rate of non-vertebral osteoporotic fractures is 11.0/1000 person-years in the population of RA patients aged 50 years and more [37]. In another large study conducted based on a USA health database on both men and women aged more than 18 years comparing 92,827 RA to 921,715 non-RA controls, the incidence rate of fracture at any of the four sites (wrist, humerus, hip, and pelvis) among RA patients was 9.6 per 1000 person-years and 1.5 times higher than that of non-RA patients (6.3 per 1000 person-years) [33].

In contrast with other peripheral sites, the incidence of wrist fracture decreased in RA as compared to the controls [6,32]. It could be due to different risk factors, especially for the mechanisms of fall. It was expected to find the same risk factors for a fracture in the RA population and the general population. Patients with incident vertebral deformities or peripheral fractures are significantly older [16,22], had lower BMD, higher disability and more often a previous fracture and a higher cumulative steroid [11,15,20], frequent falls [35], low body mass index (BMI), alcohol abuse, smoking [28], risk factors resulting from the disease itself such as the intensity of the inflammation and disability [29,36,40], duration of the disease [1,37], and cachexia [18].

Similar to the general population, patients with a morphometric, self-reported vertebral fracture or self-reported non-vertebral fracture have a higher statistically significant 10-year probability of sustaining a major osteoporotic fracture or hip fracture than patients without fractures [23].

According to Avouac et al. [9], BMD incompletely predicts patients who will suffer from an osteoporotic fracture, since only half of RA patients with fractures had BMD ≤ −2.5 SD. A higher proportion of RA patients have vertebral fractures, although their BMD is above the osteoporotic threshold. These data suggest that other parameters, especially the bone microarchitecture, have to be considered and underlines the need for other techniques and strategies to assess the risk of fracture in RA, such as the FRAXtool^®^ or the trabecular bone score (TBS). The fracture risk assessment tool^®^ developed by the World Health Organization (FRAXtool^®^) [48] computes the 10-year probability of a hip fracture or a major osteoporotic fracture. It integrates the risks associated with BMD at the femoral neck as well as clinical risk factors such as age, height, weight, sex, smoking status, alcohol drinking, use of glucocorticoids, and history of secondary osteoporosis. FRAXtool^®^ is useful to set a threshold in order to screen candidates for pharmacological intervention and the proportion of RA patients with a greater risk of osteoporotic fractures detected with FRAXtool^®^ is higher than with the WHO criteria [19]. When applied to 238 patients enrolled in the Canadian Early Arthritis Cohort (CATCH) study without using BMD measurements, FRAXtool^®^ identified 5% to 13% of patients at a high risk for fracture [49]. Patients with a higher osteoporotic fracture risk determined by FRAXtool^®^ are more likely to be female, especially postmenopausal women, with alcohol use, glucocorticoid use, proton pump inhibitor use, and had lower BMI (<25 kg/m^2^) [19]. High fracture risk patients identified by the FRAXtool^®^ was also statistically associated with disease duration, menopause duration, disease activity score (DAS28), and health assessment questionnaire (HAQ) [25].

The trabecular bone score (TBS) is also a useful, new tool to improve the prediction of fragility fracture. It is a parameter that can be automatically calculated during a BMD measurement. It assesses the quality of bone texture using the pixel gray-level variations in dual-energy X-ray absorptiometry images. A low TBS value suggests a low quality of bone texture. In the RA population, a low TBS is associated with a higher risk of having vertebral fractures even after adjusting for confounding factors such as age, DAS28, and BMD [17,24]. TBS can now be combined with the FRAXtool^®^. The TBS could be a supplementary tool for discriminating osteoporotic fractures risk in postmenopausal women with RA. It may have a nonlinear relationship with the cumulative dose of GCs, but not with the RA disease activity.

## 3. Physiopathology of Bone Loss in Rheumatoid Arthritis

As expected, classical risk factors for osteoporosis are still valuable to predict osteoporosis in patients with RA. In most of the studies, patients with spinal or femoral osteoporosis were more often female than male [3], were older [26], have a previous history of low trauma fracture, [12,17] had a lower BMI, a longer disease duration [10], a corticosteroid treatment [11], a high Larsen score [14], and a higher HAQ score [11,23,29]. There is an excess of bone loss among RA women compared to controls with post-menopausal status and HRT reduces bone resorption regardless of glucocorticoid therapy in post-menopausal RA patients [2]. There could also be an interaction between estrogen deficiency and RA-genetic risk alleles promoting enhanced Th17-cell autoreactivity, manifested by ACPA (+) RA that exacerbates the inflammatory conditions and enhances loss.

In addition to these classical risk factors, there is a crosstalk between bone cells and the immunity system and chronic inflammation that can perturb bone metabolism and promote an increase in bone loss. There is a consistent association between bone loss and the CRP reflecting disease activity [3,20,50].

The association between joint damage progression and both hand and generalized BMD loss in RA suggests common mechanisms, with hand BMD loss occurring earlier than generalized BMD loss in the disease course [51,52].

In the CATCH study, the disease activity score with 28 joint counts in the high-risk fracture group was significantly higher compared to the low-risk fracture group (*p* = 0.048) [49].

### 3.1. Role of Cytokines

Pro-inflammatory cytokines, particularly tumor necrosis factor (TNF) and interleukins IL-1, IL-6, and IL-17 play a key role in the pathogenesis of inflammation-induced bone loss. The TNF-α stimulates the receptor activator of nuclear factor kappa B ligand (RANKL), which is a membrane protein secreted by osteoblasts that binds to the RANK receptor on osteoclast precursors and induces osteoclast maturation and activation leading to bone resorption and inhibits osteoblast function thus bone formation [53]. In addition, osteoprotegerin (OPG) acts as a decoy receptor, the RANKL/OPG being of major importance to assess the balance between bone formation and resorption. OPG expression is increased in anti-TNF treated patients while RANKL is not decreased even if its level can predict the therapeutic response to anti-TNF therapy.

In mice, despite TNF-mediated inflammatory arthritis, systemic bone is fully protected by the absence of interleukin-1 (IL-1), which suggests that IL-1 is an essential mediator of inflammatory osteopenia [54]. Thus, both TNF-α and IL1 play a key role in the pathogenesis of inflammation-induced bone loss in RA patients.

The Wnt/B catenin pathway is a crucial regulatory pathway for bone formation inhibiting osteoblastic differentiation and activity. Dickkopf-1 (Dkk-1) and sclerostin (SCL) are blockers of the Wnt-signaling pathway and play an important role in RA. TNF-α increases Dkk-1 which down regulates the Wnt pathway, blocking osteoblast differentiation and inducing expression of SCL leading to the apoptosis of osteocytes.

### 3.2. Bone Remodeling Markers in RA

Histomorphometric analysis from trans-iliac bone biopsies show that the low bone mass was not only due to an increased bone resorption but also due to reduced bone formation [55]. These mechanisms of bone loss have been assessed in many studies [56] by measuring bone remodeling markers: Osteocalcin, bone alkaline phosphatase (BAP), carboxy-terminal collagen cross links (CTX), carboxy-terminal propeptides of type 1 collagen (ICTP), and deoxypyridinoline. These bone remodeling markers are surrogates to evaluate bone formation, resorption, and further risk of fractures. I-CTX, which is derived from a matrix metalloprotease activity (MMPs), reflects joint erosions while I-CTX, produced by cathepsin K, gives evidence of general osteoporosis. In the COBRA study, high levels of measured I-CTX and II-CTX were strong and independent predictors of articular damages [57].

## 4. Effects on Bone of Treatments for Rheumatoid Arthritis

The DMARDs reduce the inflammation and by the way are associated with a reduced bone loss in RA patients [40]. In some studies, BMD is decreased in RA according to corticosteroid usage, especially at the lumbar spine in both females and males with RA, and in those who are current users of steroids [1,9,11] with an increased risk of fracture [21]. The effect of corticosteroid may vary according to skeletal sites [4].

### 4.1. Corticosteroids

Other authors were not able to show a detrimental effect of corticosteroids on bone in RA patients [32,58] and some showed even a protective effect probably mediated by a better control of inflammation [35], which is known to be associated with a decrease in structural damage and bone loss [59,60].

In the Canadian Early Arthritis Cohort (CATCH), there was a significant correlation between increased fracture risk groups measured by the FRAXtool^®^ and oral glucocorticoid use (*p* = 0.012) and baseline erosions (*p* = 0.040) [49].

The use of corticosteroids may be an indicator of more severe disease and a poorer functional status if the degree of systemic inflammation and the severity of RA correlated with the risk of fracture. Their detrimental effect on the fracture might have been confounded if the drug was selectively given to patients with a higher degree of systemic inflammation and RA severity [33].

In contrast, several studies recently reported a potentially beneficial effect of low-dose, short-term systemic glucocorticoids on BMD in RA suggesting that the deleterious effect of prednisolone on bone may be counteracted by its anti-inflammatory effect [47,60,61].

Other drugs, used specifically or not for RA, have been suspected to be detrimental for bone such as opioids, selective serotonin reuptake inhibitors (SSRIs), and anticonvulsants. All were associated with an increased risk of fracture in patients with RA [37,62].

### 4.2. Biological Agents

In RA patients, BMD is inversely correlated to serum levels of TNF-α and an optimal control of inflammation has been shown to reduce bone loss in this inflammatory disease.

Some observational or randomized study designs have reported that inflammatory generalized bone loss is suppressed by biological DMARDs targeting pro-inflammatory mediators such as the TNF-α (adalimumab, certolizumab, etanercept, golimumab, and infliximab), CD80/CD86 (abatacept), IL-6 (tocilizumab), CD20 (rituximab), and IL-1 (anakinra) both at the hip and lumbar spine. Nevertheless, this global protective effect was associated with a continuing bone loss in the BMD of the hands, which could reflect a suboptimal suppression of local inflammation [63,64,65]. This effect on BMD is accompanied by a significant increase in osteocalcin serum levels and a significant decrease in markers for bone resorption [66].

## 5. Conclusions

Bone loss in RA patients is well documented and is a frequent comorbidity. Fractures resulting from osteoporosis ranked high among comorbidities in contributing to mortality, future hospitalizations, and increased disability. Regular bone mineral density (BMD) measurements and fracture risk assessments using tools such as the FRAX algorithm should be performed for an early detection of osteoporosis in RA patients. Furthermore, the VFA technology on DXA devices should be used in these patients at the time of BMD measurement. We presume that with a lesser use of corticosteroids due to a better control of the inflammation and the potential protective effect of biologic agents that the prevalence of osteoporosis will slightly decrease. However, the risk of fractures remains high in this population highlighting the need for monitoring osteoporosis among RA patients and justifying its prevention and treatment.

## Figures and Tables

**Table 1 jcm-09-03361-t001:** Bone mineral density (BMD) in rheumatoid arthritis (RA) patients: Observational studies.

Authors Country Year	Studied Population	Type of the Study	BMD		
			Sites		
			Total Hip	Femoral Neck	Lumbar Spine
Laan RFJM The Netherlands 1993 [10]	men and women (58.7%) mean age: women 57.5 95% CI 27 to 78 men 57 95% CI 30 to 80	longitudinal 97 RA	Z-score < −2 = 4.4% 95% CI 0.2 to 8.6		Z-score < −2 = 9.3% 95% CI 3.5 to 15.0
Sinigaglia L Italy 2000 [11]	women 20 to 70 years	observational 925 RA		36.2% T-score ≤ −2.5 SD	28.8% T-score ≤ −2.5 SD
Kvien TK Norway 2000 [12]	women mean age: 54.8 ± 11.6 years Oslo RA Register	observational 394 RA	14.7% T-score ≤ −2.5 SD 95% CI 11.1 to 18	14.7% 95% CI 11.1 to 18.3	16.8% 95% CI 13.1 to 20.5
Haugeberg G Norway 2000 [13]	women age: 20 to 70 years county RA register	observational 394 RA	all RA patients: 14.7% T-score ≤ −2.5 SD 95% CI 11.1 to 18.3 corticoid non-users: 5.8 T-score ≤ −2.5 SD 95% CI 1.9 to 9.7 corticoid users: 24.0 T-score ≤ −2.5 SD 95% CI 17.1 to 30.9	all RA patients: 14.7% T-score ≤ −2.5 SD 95% CI 11.1 to 18.3 corticoid non-users: 6.5 95% T-score ≤ −2.5 SD 95% CI 2.4 to 10.6 corticoid users: 24.7 T-score ≤ −2.5 SD 95% CI 17.7 to 31.7	all RA patients: 16.8% T-score ≤ −2.5 SD 95% CI 13.1 to 20.5 corticoid non-users: 8.6% T-score ≤ −2.5 SD 95% CI 3.9 to 13.3 corticoid users: 26.6% T-score ≤ −2.5 SD 95% CI 19.6 to 33.6
Lodder MC Norway, UK, The Netherlands 2003 [14]	women from OSTRA cohort mean age: 61.0 ± 5.8 years age: 50 to 70 years	observational 150 RA	8% T-score ≤ −2.5 SD	12% T-score ≤ −2.5 SD	10.7% T-score ≤ −2.5 SD
Ørstavik RE Norway 2003 [15]	women Oslo RA register mean age: 63.4 (51.4 to 73.6) years	observational 229 RA	16.6% T-score ≤ −2.5 SD	20.3% T-score ≤ −2.5 SD	21.8% T-score ≤ −2.5 SD
El Maghraoui Morocco 2010 [16]	women mean age: 49.4 ± 7.3 years age: 27 to 70 years	observational 172 RA	Any site: 44.2% T-score ≤ −2.5 SD		
Lee SG South Korea 2012 [8]	women age: 20 to 80 years	case-control 299 RA 246 non-RA controls	7.8% ** T-score ≤ −2.5 SD		18.2% T-score ≤ −2.5 SD *p* = 0.067
Brébant S France 2012 [17]	women mean age: 56.0 ± 13.5 years	observational 185 RA	21.2% T-score ≤ −2.5 SD	33.3% T-score ≤ −2.5 SD	24.2% T-score ≤ −2.5 SD
El Maghraoui Morocco 2015 [18]	men and women (82.6%) mean age: 54.1 ± 11.5 years age: 25 to 82 years	observational 178 RA	overall population: 29.2% T-score ≤ −2.5 SD		
Choi ST South Korea 2018 [19]	men and women (88.9%) mean age: 61.8 ± 11.5 years	retrospective cross-sectional study 479 RA	overall population: 91.3% T-score ≤ −2.5 SD men: 90% T-score ≤ −2 T.5 SD women: 91.3% T-score ≤ −2.5 SD		

Level of significance between RA and non-RA controls: ** *p* < 0.02; NS: Non-significant; OR: Odd ratio; RR: Relative risk; HR: Hazard ratio.

**Table 2 jcm-09-03361-t002:** BMD in RA patients: Case-control studies.

Authors Country Year (Ref)	Studied Population	Type of the Study *n* RA/ *n* Controls	BMD		
			Sites		
			Total Hip	Femoral Neck	Lumbar Spine
Garton MJ UK 1993 [1]	men age: 47 to 74 years	40 RA 20 non-RA controls	Trochanter RA corticoid users vs. non-RA: −15.9% ** RA non-users vs. non-RA: −10.1 *		RA corticoid users vs. non-RA: −12.0% ** RA non-users vs. non-RA: −11.2 **
Hall GM UK 1993 [2]	postmenopausal women	195 RA 597 non-RA controls	RA corticoid users vs. non-users: −7.4 *** 95% CI −1.2 to 13.6 RA corticoid non-users vs. non-RA: −6.9% * 95% CI 3.4 to 10.3 RA corticoid users vs. non-RA: −13.8% * 95% CI 8.6 to 19.0	RA corticoid users vs. RA non-users: −6.9 ** 95% CI −3.4 to −10.3	RA corticoid users vs. RA non-users: −6.5 * 95% CI 0 to 13.0 RA corticoid non-users vs. non-RA: NS RA corticoid users vs. non-RA: −7.5 * 95% CI 1.8 to 13.2
Gough AKS UK 1994 [3]	men and women (66.9%)	148 RA 50 non-RA controls	loss of BMD in 1-year at the trochanter all: −2.2 ± 0.5 *** male: −0.6 ± 0.6 *** female: −2.9 ± 0.6 ***	loss of BMD in 1-year all: −2.0 ± 0.4 male: −12 ± 0.7 female: −23 ± 0.6	loss of BMD in 1-year all: −1.0 ± 0.3 * male: −0.2 ± 0.5 * female: −13 ± 0.4 *
Lane NE USA 1995 [4]	non-black women ≥ 65 years SOF study	120 RA 7966 non-RA controls	RA corticoid non-users vs. non-RA: −7.3 * 95% CI −11.4 to −3.2 RA corticoid users vs. non-RA: −14.7 * 95% CI −23.4 to 5.9		RA corticoid non users vs. non-RA: −8.4 * 95% CI −13.4 to −3.4 RA corticoid users vs. non-RA: NS (too small number)
Martin JC UK 1997 [5]	caucasian postmenopausal women from EVOS	46 RA 29 non-RA controls		RA vs. non-RA: −15.4% **	RA vs. non-RA: −6.7%
Haugeberg G Norway 2000 [7]	men age: 20 to 70 years county RA register	case-control 94 RA men 1130 non-RA controls: European/United States reference population	age group 60–70 years: −6.9% *	age group 60–70 years: −5.2% *	NS
Ørstavik RE Norway 2004 [6]	women Oslo RA register mean age: 63.0 (50.7 to 73.6) years	249 RA	17.3% T-score ≤ −2.5 SD * OR = 5.8 95% CI 2.4 to 17.0	18.6% T-score ≤ −2.5 SD * OR = 4.1 95% CI 2.0 to 9.7	22.0% T-score ≤ −2.5 SD * OR = 2.0 95% CI 1.2 to 3.4
Lee SG South Korea 2012 [8]	women age: 20 to 80 years	299 RA 246 non-RA controls	7.8% ** T-score ≤ −2.5 SD		18.2% T-score ≤ −2.5 SD *p* = 0.067
Avouac J France 2012 [9]	women mean age: 61 ± 11 years	139 RA 227 non-RA controls	19% T-score ≤ −2.5 SD ***		21% T-score ≤ −2.5 SD ***

Level of significance between RA and non-RA controls: * *p* < 0.05; ** *p* < 0.02; *** *p* < 0.001; NS: Non-significant.

**Table 3 jcm-09-03361-t003:** Incidence, prevalence, and risk of fractures in RA patients: Observational studies.

Authors Country Year (Ref)	Studied Population	Type of Study *n* RA	Fractures Sites			
			Vertebral	Proximal Femur	Non-Vertebral	Any Site
Kvien TK Norway 2000 [12]	women age: 54.8 ± 11.6 years Oslo RA Register	observational 394 RA			21.2%	
Sinigaglia L Italy 2000 [11]	women 20 to 70 years	observational 925 RA	8.0%			
Lodder MC Norway UK, The Netherlands 2003 [14]	women from OSTRA cohort mean age: 61.0 ± 5.8 years age: 50 to 70 years	observational 150 RA	16%			
Ørstavik RE Norway 2002 [15]	women Oslo RA register mean age: 63.4 (51.4 to 73.6) years	observational 229 RA	48.9%			
Ørstavik RE Norway 2005 [6]	women Oslo RA register mean age: 53.0 ± 11.2 years	longitudinal 255 RA	15.0% (VFA) IR = 2.9/100 person-years			
El Maghraoui Morocco 2010 [16]	women mean age: 49.4 ± 7.3 years age: 27 to 70 years	observational 172 RA	36% (VFA)			
Vis M Norway, UK, The Netherlands 2011 [20]	women OSTRA cohort mean age: 61 years age: 20 to 70 years	Prospective 102 RA	33% IR = 3.7/100 person-year 95% CI 2.2 to 5.8		35% IR = 3.7/100 person-years 95% CI 2.2 to 5.8	
El Maghraoui Morocco 2015 [16]	men and women (82.8%) mean age: 54.1 ± 11.5 years	observational 178 RA	37% (VFA)			
Coulson KA USA 2009 [21]	women CORRONA registry	prospective 8419 RA	IR = 0.78/100 person-years	IR = 0.66/100 person-years	IR = 2.8/100 person-years	IR = 3.71/100 person-years
Brébant S France 2012 [17]	women 56.0 ± 13.5 years	observational 185 RA	17.8% (VFA)			31.3%
Dirven L The Netherlands 2012 [22]	BeSt study men and women (67%) mean age: 54 years	randomized trial 275 RA	15.0%			
El Maghraoui Morocco 2015 [18]	men and women (82.6%) mean age: 54.1 ± 11.5 years age: 25 to 82 years	observational 178 RA				
Rentero ML Spain 2015 [23]	Women age ≥ 18 years mean age: 59.6 ± 15.0 years	observational 480 RA	20.0% (VFA)		9.8%	
Choi YJ South Korea 2017 [24]	women age ≥ 50 years	observational 279 RA	12.5%			
Choi ST South Korea 2018 [19]	men and women (88.9%) age: 61.8 ± 11.5 years	retrospective cross-sectional study 479 RA	16.9%	0%	0%	
Phuan-udon R Thailand 2018 [25]	men and women (89%) Siriraj RA Cohort mean age: 61.6 ± 9.91 age: 40 to 90 years	232 RA				45.7%

Level of significance between RA and non-RA controls: ** *p* < 0.02; *** *p* < 0.001; NS: Non-significant; OR: Odd ratio; RR: Relative risk; HR: Hazard ratio; IR: Incidence rate; VFA: Vertebral fracture assessment.

**Table 4 jcm-09-03361-t004:** Incidence, prevalence, and risk of fractures in RA patients: Case-control studies.

Authors Country Year	Studied Population	Type of Study *n* RA/ *n* Controls	Fractures Sites			
			Vertebral	Proximal Femur	Non-Vertebral	Any Site
Hooyman JR USA 1984 [26]	women Rochester	population-based 388 RA	10.8%	7.5% RR = 1.51% * 95% CI 1.01 to 2.17	19.6%	30.4%
Verstraeten A Belgium 1986 [27]	postmenopausal women	case-control 104 RA 43 controls	7.7%	1.0%		
Spector UK 1993 [28]	postmenopausal women age 45 to 65 years RA: Five London hospitals controls: London practice register	case-control 149 RA 713 non-RA controls	12.1% OR = 2.1 95% CI 1.2 to 3.7			
Cooper C UK 1995 [29]	men and women (80%) age: 50 to 99 years	case-control 300 RA 600 non-RA controls		OR = 2.1 95% CI 1.0 to 4.7 *p* = 0.06 corticosteroids users: OR = 2.5 95% CI 1.1 to 5.5 *		
Peel NFA UK 1995 [30]	postmenopausal women mean age: 65 years (50 to 79) years	case-control 76 RA corticosteroids users 347 non-RA controls	27.6% OR = 6.2 95% CI 3.2 to 12.3			
Martin JC UK 1997 [5]	postmenopausal women from EVOS	case-control 46 RA 29 non-RA controls	corticosteroids users: 20% non-corticosteroids users: 23.8%			
Huusko TM Finland 2001 [31]	men and women (72%) mean age: Men 79 (62 to 92) women 77 (52 to 91)	case-control fractures Non-RA controls with hip fractures		OR = 3.26 95% CI 2.26 to 4.70		
Ørstavik RE Norway 2004 [6]	women Oslo RA register mean age: 63.0 (50.7 to 73.6) years	case-control 249 RA 249 non-RA controls	59.0%			
Van Staa TP UK 2006 [32]	General Practice Research Database men and women (71.1%) ≥ 40 years	case-control 30,262 RA 90,783 controls	RR = 2.4 95% CI 2.0 to 2.8	RR = 2.0 95% CI 1.8 to 2.3		RR = 1.5 95% CI 1.4 to 1.6
Kim SY USA 2010 [33]	men and women (73%) Health Core Integrated Research Database age ≥ 18 years median age: 55 years	case-control 92,827 RA 921,715 non-RA controls		men IR = 2.4/1000 person-years women IR = 3.8/1000 person-years HR = 1.44, 95% CI 1.24 to 1.67		1.4% IR = 9.6/1000 person-years HR = 1.26 95% CI 1.15 to 1.38 corticosteroids users: HR = 1.15 95% CI 1.03 to 1.27
Wright NC USA 2011 [34]	postmenopausal women 50 to 79 years WHI study	prospective 960 RA 83,295 non-RA controls	4.0% *** HR = 1.93 *** 95% CI 1.29 to 2.90	4.0%*** IR = 0.51/100 person-years HR = 3.03 2.03 to 4.51 ***		24.8% *** IR = 3.64/100 patient-years 95% CI 3.17 to 4.11 HR = 1.49 95% CI 1.26 to 1.75 ***
Ghazi M France 2012 [35]	women age 56.1 ± 14.2 years	case-control 101 RA 303 non-RA controls	21.8% (VFA) *** OR = 6.5 95% CI 3.1 to 13.9		28.71%	
Avouac J France 2012 [9]	Women mean age: 61 ± 11 years	case-control 139 RA 227 non-RA controls	19.0% ***		22.0% **	33.0% ***
Filho JCA Brazil 2013 [36]	men age 51.6 ± 9.3 years	case-control 50 RA 52 non-RA controls	36.0% *		4%	
Roussy JP Canada 2013 [37]	men and women Quebec healthcare databases age ≥ 50 years	case-control 27,076 RA non-RA controls		2.4%	5.6% IR = 11.0/1000 person-years 95% CI 10.4 to 11.5	
Brennan SL Australia 2014 [38]	women age ≥ 35 years database: Barwon Statistical Division	case-control 1008 RA 172,422 non-RA controls	31.7% ***	7.3% ***		1.9% IR = 114/10000 person-years RR = 1.43 95% CI 0.98–2.09 *p* = 0.08
Xue A-L USA, UK, Sweden, Norway, Finland, Australia, China 2017 [39]	men and women	meta-analysis (13 studies)	RR = 2.93 95% CI 2.25 to 3.83	RR = 2.41 95% CI 1.83 to 3.17		RR = 2.25 95% CI 1.76 to 2.87
Clynes MA UK 2019 [40]	men and women (70.1%) UK Biobank: Hospital Episode Statistics (HES)	case-control 5492 RA 497,051 non-RA controls				men: 5.1% OR = 1.46 95% CI 1.17 to 1.81 *** women: 5.5% OR = 1.71 95% CI 1.50 to 1.94 ***
Hong Taiwan 2019 [41]	men and women Nationwide database age ≥ 40 years	retrospective 30.507 RA	HR = 1.47 *** 95% CI 1.19 to 1.81			
Weiss RJ Sweden 2019 [42]	men and women (66%) Swedish National Hospital Discharge Register median age: 71 years	case-control 3379 RA 420,331 non-RA controls	OI = 2.7 * 95% CI 2.1 to 3.4	OI = 2.9 * 95% CI 2.7 to 3.1		OI = 2.9 * 95% CI 2.8 to 3.1

Level of significance between RA and non-RA controls: * *p* < 0.05; ** *p* < 0.02; *** *p* < 0.001; NS: Non- significant; OR: Odd ratio; RR: Relative risk; HR: Hazard ratio; IR: Incidence rate.

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
