# Peer review of "Bone Loss, Osteoporosis, and Fractures in Patients with Rheumatoid Arthritis: A Review"

_jcm, 2020, doi:10.3390/jcm9103361_

Round 1
Reviewer 1 Report
Minor comments:
Line 16-19: Statement too long to understand. Please simplify.
Line 28: it should be "...Last three decades"
Table 1: Please describe in text what is Z-score and how it is calculated
Line 111: Please add space between 2.5SD
Line 116: Please describe the accuracy of FRAXtool
Line 122: Please add space between "..to13%"
Line 129: citation required for TBS
Line 157,163,170: Citation required to support the scientific statements
Table 3: Sinigaglia L et al. woman age?

Author Response
Thank you for having reviewed our manuscript. Please find thereafter the required modifications:
Line 16-19: Statement too long to understand. Please simplify.
We made the sentence shorter and changed it in: “Compared with matched population, RA patients with fractures have more common risk factors of osteoporosis and fragility fractures but also risk factors resulting from the disease itself like duration, intensity of the inflammation and disability, and cachexia. »
Line 28: it should be "...Last three decades"
Changed
Table 1: Please describe in text what is Z-score and how it is calculated
We added this explanation in the text: “Results are expressed whether in T-score or Z-score. T-score is a young gender-matched population at peak bone mass, while the Z-score is derived from an age-matched reference population. Both T-scores and Z-scores are derived by comparison to the reference population on a standard deviation scale.”
Line 111: Please add space between 2.5SD
Modified
Line 116: Please describe the accuracy of FRAXtool
After having given explanations for how FRAXtoolÒ was calculated we inserted in line 124 the following referenced sentence: … and the proportion of RA patients with a greater risk of osteoporotic fractures detected with FRAXtool is higher than with WHO criteria [19].
Line 122: Please add space between "..to13%"
Modified
Line 129: citation required for TBS
TBS is already cited in the following references:
(17) Bréban, S.; Briot, K.; Kolta, S.; Paternotte, S.; Ghazi, M.; Fechtenbaum, J.; Roux, C. Identification of Rheumatoid Arthritis Patients With Vertebral Fractures Using Bone Mineral Density and Trabecular Bone Score. J. Clin. Densitom. 2012, 15 (3), 260–266. https://doi.org/10.1016/j.jocd.2012.01.007.
(24) Choi, Y. J.; Chung, Y.-S.; Suh, C.-H.; Jung, J.-Y.; Kim, H.-A. Trabecular Bone Score as a Supplementary Tool for the Discrimination of Osteoporotic Fractures in Postmenopausal Women with Rheumatoid Arthritis. Medicine (Baltimore) 2017, 96(45), e8661. https://doi.org/10.1097/MD.0000000000008661.
Line 157,163,170: Citation required to support the scientific statements
The scientific statements on the role of cytokines is given in the following citation:
(53) Redlich, K.; Smolen, J. S. Inflammatory Bone Loss: Pathogenesis and Therapeutic Intervention. Nat. Rev. Drug Discov.2012, 11 (3), 234–250. https://doi.org/10.1038/nrd3669.
Table 3: Sinigaglia L et al. woman age? We added the age range: “20 to70 yrs”

Reviewer 2 Report
The authors did a good job of reviewing the risk of osteoporosis in RA patients.
There are some relevant important topic areas missing to complete the review; please discuss effects of hormones on bones, particularly in post-menopausal women, especially in light of RA being more common in females.
It needs a section on recommendations, the prevention and management of osteoporosis in RA to complete the review.
Regarding effects of RA therapy on bone, please discuss the effetcs of conventional DMARDs - are there any effects?
Please discuss the consequence and complications as a result of increased fractures due to bone loss. Comment on surgery and related cost incurred.
There are scattered sentences throughout the mansucript that could do with updating the grammar, perhaps proof-reading by someone who is a native English speaker may help.
Author Response
Thank you for having reviewed our manuscript. Please find thereafter the required modifications:
Line 151: we added a reference on this topic: There is an excess of bone loss among RA women compared to controls with post-menopausal status and HRT reduces bone resorption regardless of glucocorticoid therapy in post-menopausal RA patients [2]. There could be also an interaction between estrogen deficiency and RA-genetic risk alleles promoting enhanced Th17- cell autoreactivity, manifested by ACPA(+) RA that exacerbates the inflammatory conditions and enhances loss.
It needs a section on recommendations, the prevention and management of osteoporosis in RA to complete the review.
In the conclusion we provide recommendations: “Regular bone mineral density (BMD) measurement and fracture risk assessment using tools such as FRAX algorithm should be performed for early detection of osteoporosis in RA patients. Furthermore, VFA technology on DXA devices should be used in these patients at the time of BMD measurement. We presume than with a lesser use of corticosteroids due to a better control of the inflammation and the potential protective effect of biologic agents that the prevalence of osteoporosis will slightly decrease. But the risk of fractures remains high in this population highlighting the need for a monitoring of osteoporosis among RA patients and justifying its prevention and treatment. “
Regarding effects of RA therapy on bone, please discuss the effects of conventional DMARDs - are there any effects?
We added in chapter 4. Effects on bone of treatments for rheumatoid arthritis the following sentence and reference: The DMARDs reduces the inflammation and by the way are associated with a reduced bone loss in RA patients [40].
Please discuss the consequence and complications as a result of increased fractures due to bone loss. Comment on surgery and related cost incurred.
Unfortunately, we did not find any publication about mortality and costs induced by osteoporosis and fractures in RA populations.
There are scattered sentences throughout the manuscript that could do with updating the grammar, perhaps proof-reading by someone who is a native English speaker may help.

Reviewer 3 Report
This is a well written review regarding osteoporosis and risk of fractures in patients with rheumatoid arthritis. I would just recommend the authors to describe the literature search methodology for making Tables 1, 2, 3 and 4, which makes sure that this review was written in an unbiased fashion.
Author Response
Thank you for having reviewed our manuscript. Please find thereafter the required modifications:
The relevant papers were searched through Pubmed database with the following key words: rheumatoid arthritis, osteoporosis, fractures, bone, bone mineral density, bone remodeling markers.

Round 2
Reviewer 2 Report
Thank you for the revision, the additions have improved the manuscript. The grammar and language still needs updating, as previously suggested, could do with proof reading by a native English speaker.